# DirectTriGS: Triplane-based Gaussian Splatting Field Representation for 3D Generation

## Abstract

We present DirectTriGS, a novel framework designed for 3D object generation with Gaussian Splatting (GS). GS-based rendering for 3D content has gained considerable attention recently. However, there has been limited exploration in directly generating 3D Gaussians compared to traditional generative modeling approaches. The main challenge lies in the complex data structure of GS represented by discrete point clouds with multiple channels. To overcome this challenge, we propose employing the triplane representation, which allows us to represent Gaussian Splatting as an image-like continuous field. This representation effectively encodes both the geometry and texture information, enabling smooth transformation back to Gaussian point clouds and rendering into images by a TriRenderer, with only 2D supervisions. The proposed TriRenderer is fully differentiable, so that the rendering loss can supervise both texture and geometry encoding. Furthermore, the triplane representation can be compressed using a Variational Autoencoder (VAE), which can subsequently be utilized in latent diffusion to generate 3D objects. The experiments demonstrate that the proposed generation framework can produce high-quality 3D object geometry and rendering results.

## 1 Introduction

Neural rendering has grown to a focal point in rendering techniques in recent years, as it achieves more realistic rendering effects by leveraging the great expressiveness of neural network. The representatives are neural radience field (NeRF) (Mildenhall et al., 2021) and the newly emerged Gaussian Splatting (GS) (Kerbl et al., 2023). However, when applied to the field of 3D generation, the slow rendering and training speed become a strong limitation for NeRF. While GS is more flexible with greater rendering efficiency and editability, few works have addressed the challenge of direct GS generation in 3D due to its complex data structure. In this paper, we propose a novel approach called DirectTriGS, which encodes GS as triplane representation, and introduce its corresponding renderer, TriRenderer. Subsequently, we apply latent diffusion on the Triplane representation to generate high-quality GS objects.

Gaussian splatting uses multi-channel point cloud of "splats" to describe the 3D contents. With a differentiable splats rasterizer, GS has fast rendering speed. However, the sparsity, multiple channels, and uneven distributed density of 3D GS bring great difficulty for 3D generation. We propose to encode GS contents as multi-channel triplane representations, which have been shown to possess favorable properties for representing 3D geometry or NeRF, as demonstrated in previous works Wu et al. (2024); Shue et al. (2023). In our work, we leverage the triplane representation to encode both the geometry information and other GS attributes in two separate groups of channels. By training on a dataset of 3D objects, we obtain a shared TriRenderer that is capable of decoding different triplane representations to GS and then render it to image. TriRenderer is fully differentiable, enabling the use of only 2D rendering loss to supervise both the texture and geometry of 3D GS. The advantage of using Triplane to represent GS are two folds. First, it leads to high memory efficiency compared with dense voxels and is expressive enough for generating various 3D GS. Second, Triplane is more compatible with the convolution-based encoders compared with the original sparse GS point cloud, which require specifically designed networks on processing sparse point clouds.

We follow stable diffusion (Rombach et al., 2022) to train DirectTriGS on the proposed triplane representation. Specifically, a VAE is designed to further convert the Triplanes into latent code. We

use two separate decoders to decouple the decoding of geometry and GS attributes. Second, we roll out the triplane latent to an expanded multi-channel image, then exploit staged latent diffusion to do generation. Two-stage diffusion is employed to generate the geometry and the corresponding GS appearance. Finally, score distillation sampling (SDS) (Poole et al., 2022) is adopted as a optional post-processing to refine or restyle the generated 3D objects.

Our contribution can summarized as follows. 1) We propose a Triplane representation for direct 3D GS generation, which is memory efficient to derive GS point clouds with vivid rendering. 2) We design a fully differentiable TriRenderer to enable the end-to-end training of triplane representations, with only 2D supervisions. 3) We develop a Triplane-based GS generation framework DirectTriGS that incorporates a specially designed VAE, latent diffusion module, and a SDS based refiner. 4) Experiments demonstrate our method produces competitive performance in both 3D geometry and multi-view rendering quality.

## 2 RELATED WORKS

**Gaussian Splatting.** Gaussian Splatting (Kerbl et al., 2023) exploits pointcloud of splats to describe the 3D content, and every splat is a 3D ball with a shape of Gaussian distribution, and it has other properties like opacity, color or SH parameters. The GS can be rendered like mesh using a special designed rasteraization, which allows real-time rendering of photorealistic scenes. As a newly developed rendering technique, there emerges so many research focusing on its reconstruction quality enhancement and algorithm optimization (Yu et al., 2023; Cheng et al., 2024; Fan et al., 2023; Lu et al., 2023). Another related active area includes 4D dynamic GS modeling (Wu et al., 2023a; Yang et al., 2023; Chen et al., 2023c) , scene editing (Fang et al., 2023; Chen et al., 2023b) and its applications like SLAM (Matsuki et al., 2023; Yan et al., 2023).

**3D Object Generation.** Recently, there emerges many works aiming to solve the geometry generation and rendering in combination, which can be roughly divide into 2 groups. First is the route of multi-view image to 3D, which generate multi-view colored 3D image and then reconstruct the 3D shape and project image to texture, such as Poole et al. (2022); Shi et al. (2023); Höllein et al. (2023); Liu et al. (2023); Chung et al. (2023); Tang et al. (2023). Since these approaches are actually 2D generator, it is difficult to maintain 3D consistency, which may lead to Janus problem. And the second group mainly operate in 3D, which can be clustered by different 3D representation. For example, Gupta et al. (2023); Gao et al. (2022) are able to generate high-quality 3D textured meshes using differentiable rendering; Nichol et al. (2022); Wu et al. (2023c) generate colored pointcloud; Chen et al. (2023a); Metzer et al. (2023) generate NeRF volumes; Ju et al. (2023); Wu et al. (2024)generate SDF volumes.

As for GS generation, some works exploit multi-view image based route such as Chung et al. (2023); Tang et al. (2023), where the image generator output image from required views to enhance the GS rendering in a recursive manner. Rare works explore the GS generation directly in 3D space, such as Zou et al. (2023) generates a point cloud using a Transformer structure with 2D image as input, and build a mapping of image to tri-plane encoding of GS attribute; GaussianCube (Zhang et al., 2024) proposes to use optimal transport to model the 3D GS for text-to-3D generation.

## 3 REVISITING GAUSSIAN SPLATTING

Gaussian Splatting Kerbl et al. (2023) uses a set of Gaussian points to describe a 3D object. The Gaussian points are defined by a full 3D covariance matrix $\Sigma$ in a world space as $G(\mathbf{x}) = e^{-\frac{1}{2}\mathbf{x}^T \Sigma^{-1} \mathbf{x}}$, centered at point (mean) $\mathbf{p} \in \mathbb{R}^3$. Each point is with a opacity scalar $\alpha \in [0, 1]$ for blending and a series of Spherical Harmonics (SH) coefficients to correctly capture the view-dependent appearance of the scene. The number of SH coefficients are $3 \times (n+1)^2$, where $n$ denotes the SH order, and higher $n$ corresponds to more accurate view-dependent appearance.

During rendering, the Gaussians are project to 2D given a viewing transformation $\mathbf{W}$. The covariance is transformed as $\Sigma' = \mathbf{J}\mathbf{W}\Sigma\mathbf{W}^T\mathbf{J}^T$, where $\mathbf{J}$ is the Jacobian of the affine approximation of the projective transformation. To ensure $\Sigma$ to be semi-positive in the whole training process, $\Sigma$ can be represented by a Cholesky decomposition as $\Sigma = \mathbf{R}\mathbf{S}\mathbf{S}^T\mathbf{R}^T$, represented by a tuple $s = (s_1, s_2, s_3)$ for scaling and a unnormalized quaternion $\mathbf{q} \in \mathbb{R}^4$ for rotation. In summary, each Gaussian point are

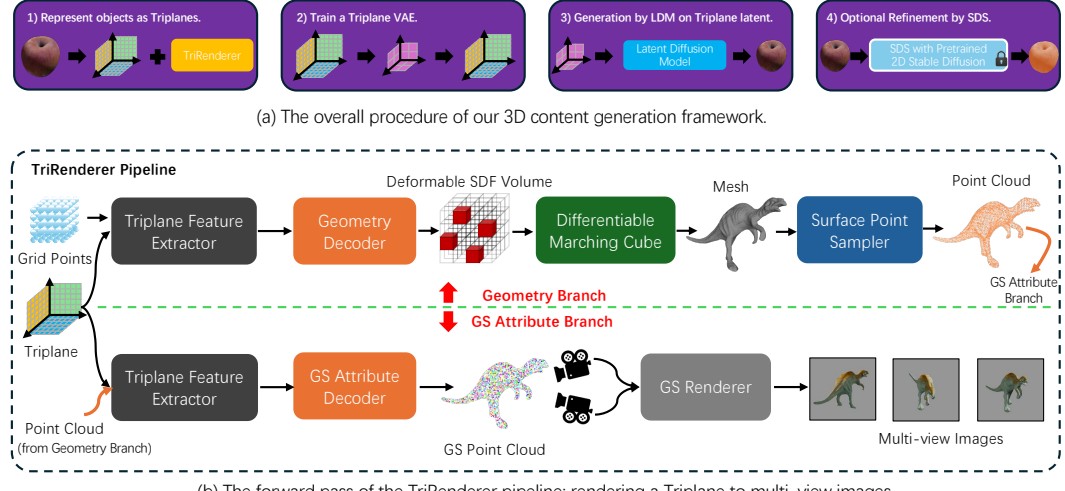

(a) The overall procedure of our 3D content generation framework.

(b) The forward pass of the TriRenderer pipeline: rendering a Triplane to multi-view images.

Figure 1: The overview of the proposed 3D GS generation framework and the core component TriRenderer. We use triplane to represent 3D objects, and design a TriRenderer to decode triplane to GS pointcloud and render multi-view images.

a union set of position $\mathbf{p}$ and its GS attributes:

$$\mathcal{G}_{\mathbf{p}} := \{s, \mathbf{q}, \text{SH}, \alpha\}. \tag{1}$$

GS pointcloud can be efficiently rendered by a rasterization-based splatting renderer (Kerbl et al., 2023), which is fully differentiable.

# 4 METHODOLOGY

Our motivation is to directly generate 3D GS with fast speed by utilizing the proposed triplane-based GS representation. While 2D-lifting methods like SDS-based approaches yield intricate results, they are time-intensive and susceptible to Janus problem. Direct generation, notably with fast sampling, can complete 3D generation in under a minute, contrasting with SDS methods that take over 10 minutes or longer, such as Chen et al. (2024)Wang et al. (2024), posing challenges for users. Therefore, direct 3D generation is more practical, with 2D-lifting reserved for enhancing texture details when needed.

The procedures of our generation framework can be divide into 4 stages as shown in Fig.1 (a). Firstly, we encode the 3D objects into triplanes using the TriRenderer. The triplane encoding and TriRenderer training can be concurrently executed with only the multi-view RGBA image and camera poses as supervision. Secondly, a VAE is trained to compress the triplanes to latent space, enabling effective capture of high-level information for subsequent modules. Thirdly, a diffusion model is trained on the triplane latent code. During inference, the diffusion model produces the latent code, which can be decoded back to its corresponding triplane by the VAE decoder. Subsequently, the TriRenderer decodes the triplane into the standard GS point cloud. Finally, an SDS-based refiner can be utilized as an optional post-processing step to enhance or restyle the generated 3D object. Leveraging the pretrained 2D diffusion model within the SDS, the generated GS can be further improved to achieve a more detailed appearance.

## 4.1 TRIPLANE-BASED GS FIELD

We employ triplanes to convert the discrete multi-channel GS point cloud into a continuous field, tailored for more efficient encoding in the subsequent generative model. Triplanes are 3 planes formed by every 2 axes of $x, y, z$ axes in 3D space, where every 3D point $\mathbf{p}$ can query these planes to get corresponding features $F_{\mathbf{p}} = (F_{xy}, F_{xz}, F_{yz})$ using orthogonal projection. In real implementation, triplane is a tensor with the size of $3 \times H \times H \times C$, where $H$ denote the resolution along every axis

and $C$ is the feature channel. The query of point with arbitrary continuous coordinate should be a bi-linear intepolation in the triplane grids. Considering GS is a set of points with multiple channels of attributes, it is natural to use triplane for GS encoding. If the triplane feature and GS attributes of a point can be transformed to each other, the sparse GS pointcloud of every object can be represented as continuous triplane GS field for further encoding. We use separate channels to encode 3D geometry and other GS attributes as described in Eq. 1, which is an experimental setting for better convergence in the training.

## 4.2 TRIRENDERER

The TriRenderer serves as the crucial component for converting a 3D object into a triplane-based GS field. It acts as a fully differentiable bridge connecting the GS field with rendering, enabling the optimization from 2D images to geometry and appearance encoding.

As depicted in Fig. 1 (b), the TriRenderer comprises a geometry branch and a GS attribute branch, each equipped with independent decoders to decode the triplane. The geometry branch is responsible for retrieve the geometry as triangular mesh from triplane, with a surface sampler for GS pointcloud sampling. Then GS attribute branch uses the obtained pointcloud to query the triplane to obtain the GS attribute corresponding to each point. In this way, the GS pointcloud in the original format is retrieved, and can be rendered using the original GS renderer. It is worth mentioning that all triplanes of different object share a common TriRenderer, which ensures the features on different triplanes subject to a similar distribution.

**Geometry Branch.** We use signed distance function (SDF) to represent the geometry, and allow every vertex deform as Shen et al. (2021), so that the geometry branch decodes $F_p$ to a SDF volume and its vertice deformation. By query all grid coordinates of the designated resolution $L \times L \times L$, the deformable SDF volume is reconstructed. Then we exploit a differentiable Marching Cube algorithm FlexiCubes (Shen et al., 2023) to extract triangular mesh from the SDF volume.

**Surface Sampling.** Considering that GS pointcloud is generally gathered on the surface of objects, we randomly sample GS points on the faces of triangular mesh with barycentric coordinates. We expect that every splat is flat and with a normal consistent with its source triangular face. Assume a face is formed by vertices $< v_a, v_b, v_c >$, and vector $\mathbf{v}_1 = (v_a - v_b)$, $\mathbf{v}_2 = (v_b - v_c)$, the face normal can be calculated as $\mathbf{n} = \mathbf{v}_1 \times \mathbf{v}_2$. Then the rotation matrix of every splat drawn from this face can be formulated as

$$\mathbf{R} = [\mathbf{v}_1, \mathbf{n} \times \mathbf{v}_1, \mathbf{n}], \tag{2}$$

which can be transform to $\mathbf{q}$ in Eq.1 by standard matrix-to-quaternion algorithm. To make the splats flat, $s_3$ in Eq.1 is fixed to an infinite small value.

To further reduce the computational load in the subsequent GS rendering process, only faces oriented toward the camera are sampled. These face indices can be obtained through fast rasterization of the mesh faces. Note this rasterization is not required to be differentiable, and we use the library Nvdiffrast (Laine et al., 2020) for implementation.

**GS Attribute Branch.** By querying the triplane GS channels with the sampled pointcloud, we can obtain the GS feature and use the GS attribute branch to transform them to the rest GS attributes. Then the GS renderer can render GS pointcloud to multi-view images. Since different GS attributes have different numerical scales and distributions, we customize individual headers to decode them respectively, which is similar to Zou et al. (2023).

**Training.** The training consists of 2 stages. In the first stage, a small batch of data is used to train the triplanes and the TriRenderer together. The second stage involves the whole dataset to train all triplanes with the parameters of TriRenderer frozen. In this way, we can handle large dataset by distributed training in the second stage, and the same TriRenderer can be shared by all triplanes. There is no need to pre-train the GS pointcloud using original GS training as Kerbl et al. (2023).

The only required supervision data is $N$ multi-view images $\{I_i\}_{i=1}^N$ with camera poses, and the training losses consists of the rendering loss $L_{\text{render}}$ and the geometric regularization $L_{\text{geo}}$. The rendering loss is a weighted sum as

$$L_{\text{render}} = w_1 L_{\text{alpha}} + w_2 L_{\text{rgb}} + w_3 L_{\text{pips}}, \tag{3}$$

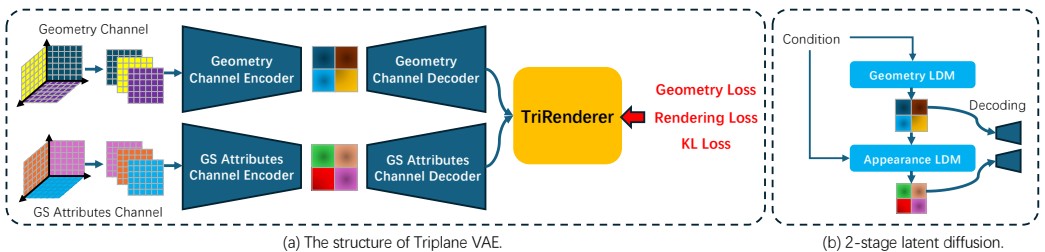

Figure 2: Triplane VAE and 2-stage diffusion.

where $L_{\text{alpha}}$ is a $L_1$ loss on the alpha map, namely the silhouette image loss; $L_{\text{rgb}}$ is a combination of $L_1$ pixel loss and SSIM loss (Wang et al., 2004) between rendered image $\hat{I}_i$ and ground truth $I_i$:

$$L_{\text{rgb}} = \sum_{i=1}^{N}(1-\beta)\|I_i - \hat{I}_i\|_1 + \beta\text{SSIM}(I_i, \hat{I}_i), \tag{4}$$

and $L_{\text{pips}}$ is the perceptual loss (Johnson et al., 2016). $w_1$, $w_2$, $w_3$ and $\beta$ are all weighting factors determined by experiments.

The geometric regularization loss on the SDF volume $V$ is defined as

$$L_{\text{geo}} = \gamma_1 L_{\text{dev}} + \gamma_2 L_{\text{weighting}} + \gamma_3 L_{\text{CE}} + \gamma_4 L_{\text{sign}}, \tag{5}$$

where the $L_{\text{dev}}$ and $L_{\text{weighting}}$ are defined by Flexicubes (Shen et al., 2023) to regularize the extracted connectivity and the weighting scale of SDF vertices. To penalize the sign changes on all grid edges, we follow Munkberg et al. (2022) to define $L_{\text{CE}}$ as

$$L_{\text{CE}} = \sum_{(s_a, s_b) \in \epsilon} \text{CE}(\sigma(s_a), \text{sign}(s_b)), \tag{6}$$

where CE denotes cross entropy; $\epsilon$ is the set of all edges connecting vertices with different signs; and $\sigma$ is the sigmoid function. Finally, $L_{\text{sign}}$ is designed to prevent the SDF volume from being trapped in an empty shape, i.e. a fully positive or fully negative SDF volume,

$$L_{\text{sign}} = \delta(V)M(|V|), \tag{7}$$

where $\delta(V) = 1$ if $V$ is empty, otherwise $\delta(V) = 0$. $M(|V|)$ denotes the mean of the absolute value of $V$.

## 4.3 TRIPLANE VAE

We employ an UNet-like structure of VAE to compress the triplane to latent space. The training pipeline is demonstrated in Fig. 2 (a). Considering that the 3 planes of triplanes encodes features from 3 orthogonal view direction, these 3 planes should be homogeneous data, so that We reshape the triplanes of batchsize $B$ to a new batch as $B \times 3 \times H \times H \times C$ to $3B \times H \times H \times C$ for the VAE training. We use decoupled encoders and decoders for geometry and GS channels of triplanes, for experiments prove that a mixed encoding may lead to blur of the rendering results. After the triplane is reconstructed, the TriRenderer trained in Section.4.1 can be used to retrieve the GS pointclouds and render it to images.

In the training process, same loss functions as Eq. 3 and Eq. 5 are used, along with an $L_1$ loss between the input and reconstructed triplanes, denoted as $L_{\text{Tri}}$. Additionally, a Kullback–Leibler divergence loss $L_{\text{KL}}$ is included to ensure that the latent variables do not deviate significantly from a normal distribution. The total losses is summarized as

$$L_{\text{VAE}} = L_{\text{Tri}} + L_{\text{Geo}} + L_{\text{Render}} + \gamma L_{\text{KL}}, \tag{8}$$

where $\gamma$ is a small weight.

Figure 3: Score distillation process(SDS) for optional texture refinement.

### 4.4 Two-stage Latent Diffusion for Text-to-3D Generation

Taking into account the significant relevance of texture appearance to the underlying geometry, we proposed to utilize staged diffusion to generate geometry and GS attribute successively as shown in Fig. 2 (b). It is easy to implement it because the latent codes for geometry and other GS attributes are totally decoupled by VAE, as mentioned earlier . In this way, the geometry code can be a new condition for the second stage.

We follow DDPM Ho et al. (2020); Karras et al. (2022) to implement the latent diffusion conditioned on the text description. To better capture the relation between different planes, we roll out the triplane latent of $B \times 3 \times h \times h \times c$ to a image-like $B \times 3h \times h \times c$ as the input of diffusion model, and the generated result will be transformed back to the original shape for TriRenderer decoding.

### 4.5 SDS-based Texture Booster

To refine or restyle the generated GS pointcloud, we utilize a pretrained 2D diffusion model for the SDS training process (Poole et al., 2022) as shown in Fig. 3. As the generated 3D GS is constrained by the mesh reconstructed from SDF, detailed in Section 4.2, we maintain the GS splats' adherence as previously established. However, we enable the mesh vertices to shift within a limited radius to enhance the stability of the SDS process. Additionally, throughout the entire SDS procedure, there is no need for densification or pruning operations. This ensures that the number of points can be controlled to align with the number of mesh faces.

With the RGB image rendered from the initial GS pointcloud, the 2D diffusion model can produce images with better 2D/3D consistence in different views, so that the optimization process can converge rapidly without Janus problem.

## 5 Experiment

The experiments are conducted step-by-step according to the generation pipeline proposed as Fig.1 (a). We use OmniObject3D (Wu et al., 2023b) dataset for toy experiments, and Objaverse (Deitke et al., 2023) dataset for main evaluation. First, we sample 200K object to train their cooresponding triplanes and TriRenderer. Then we train the VAE and diffusion model for 3D GS generation. Finally, we exploit SDS to boost the texture quality for unsatisfactory generated objects. The procedure of data pre-processing and the implementation details are included in the appendix section A.

The experimental results primarily consist of the following components: 1) A simple check of the trained triplane and VAE reconstruction. 2) For examining the proposed triplane modeling for GS generation, we focus on the text-to-3D task and showcase the direct GS generation outcomes qualitatively and quantitatively. These results are compared with Shap-E (Jun & Nichol, 2023), Direct3D Liu et al. (2024), and the most recent GaussianCube (Zhang et al., 2024). 3) In comparison to 2D-lifting GS generation methods, we present the performance of SDS refinement on unsatisfying samples, with GSGEN (Chen et al., 2024) and GaussianDreamer (Yi et al., 2024) as baselines.

### 5.1 Triplane Fitting and VAE Reconstruction

**Triplane Fitting.** We adopt a resolution of $3 \times 128 \times 128 \times 16$ for all triplane fitting, with a resolution of $64 \times 64 \times 64$ for the deformable SDF grids. We randomly sample 1000 objects for the shared TriRenderer training, and train all residual objects distributively. Every triplane is initialized with Gaussian noise before training. Under such settings, every triplane takes less than 30 seconds for geometry and GS appearance reconstruction.

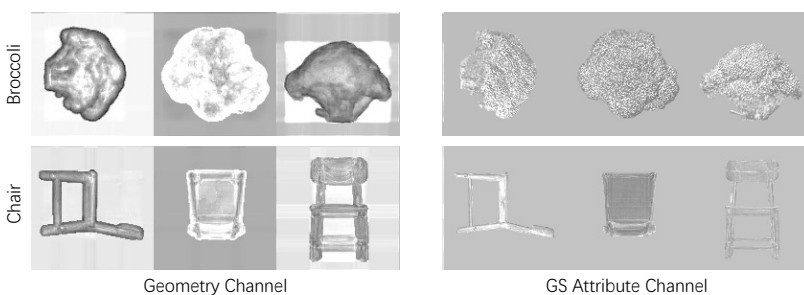

Figure 4: Channel visualization of sample triplanes.

To better investigate whether it is reasonable to encode triplane using convolution-based methods, we simply scale the channel value of trained triplanes to pixel range and visualize them as shown in Fig. 4, where clear shapes from 3 different views can be observed.

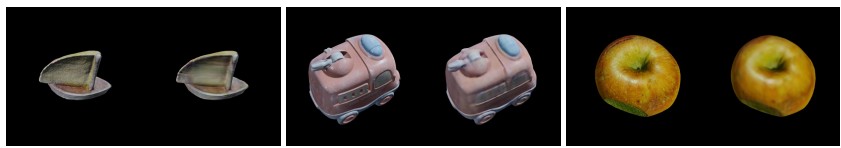

Figure 5: Triplane reconstructed by VAE. Left: ground truth. Right: reconstruction.

**VAE Reconstruction.** We use a down-sample factor of 4 to compress the triplanes to latent space. A slight blur in the reconstructed pictures are observed as Fig.5, which is inevitable but acceptable.

## 5.2 Direct 3D GS Generation by LDM

In this section, we present the results of our 3D GS object generation without the SDS refinement, both quantitatively and qualitatively. Due to the current scarcity of work targeting direct 3DGS generation, we selected two state-of-the-art text-to-3D works based on other 3D representations(NeRF) Shap-E (Jun & Nichol, 2023) and Direct3D (Liu et al., 2024) for comparison. For fairness, these selected methods also perform direct generation in the 3D domain without any refinement based on 2D diffusion. Additionally, we provide a qualitative comparison with GaussianCube (Zhang et al., 2024) in Appendix A.3.

**Qualitative Results.** The generation results from different methods are listed in Fig. 6. Every generated sample is rendered from different views accompanied by the provided text prompt captioned beneath the images. Our method showcase enhanced proficiency in both geometry and rendering quality, resulting in sharper and clearer outputs, which can be further verified in the subsequent quantitative evaluation. For more generated samples, please check the appendix section A.

**Quantitative Results.** We use CLIP score to evaluate the text-to-3d consistency, and an user study is conducted to evaluate the generation results from various aspect such as geometry, texture, realistic rendering and the consistency with given prompt. 49 users participated in the user study to score the over 50 3D samples from 1 to 5 points, and the average results are shown in Tab. 1. As for the CLIP score, the open-source repository `t2vmetrics` (Lin et al., 2024) is used to calculate the CLIP score on two versions of ViT models, and the results are demonstrated in Tab. 2. Both the CLIP score and the user study indicate that the proposed method produces better performance.

## 5.3 Texture Boosting by SDS

To compare with other SDS-based 3D GS generation methods, we implement a version of SDS for our mesh-binding GS representation as described in Section 4.5. GSGEN (Chen et al., 2024) and

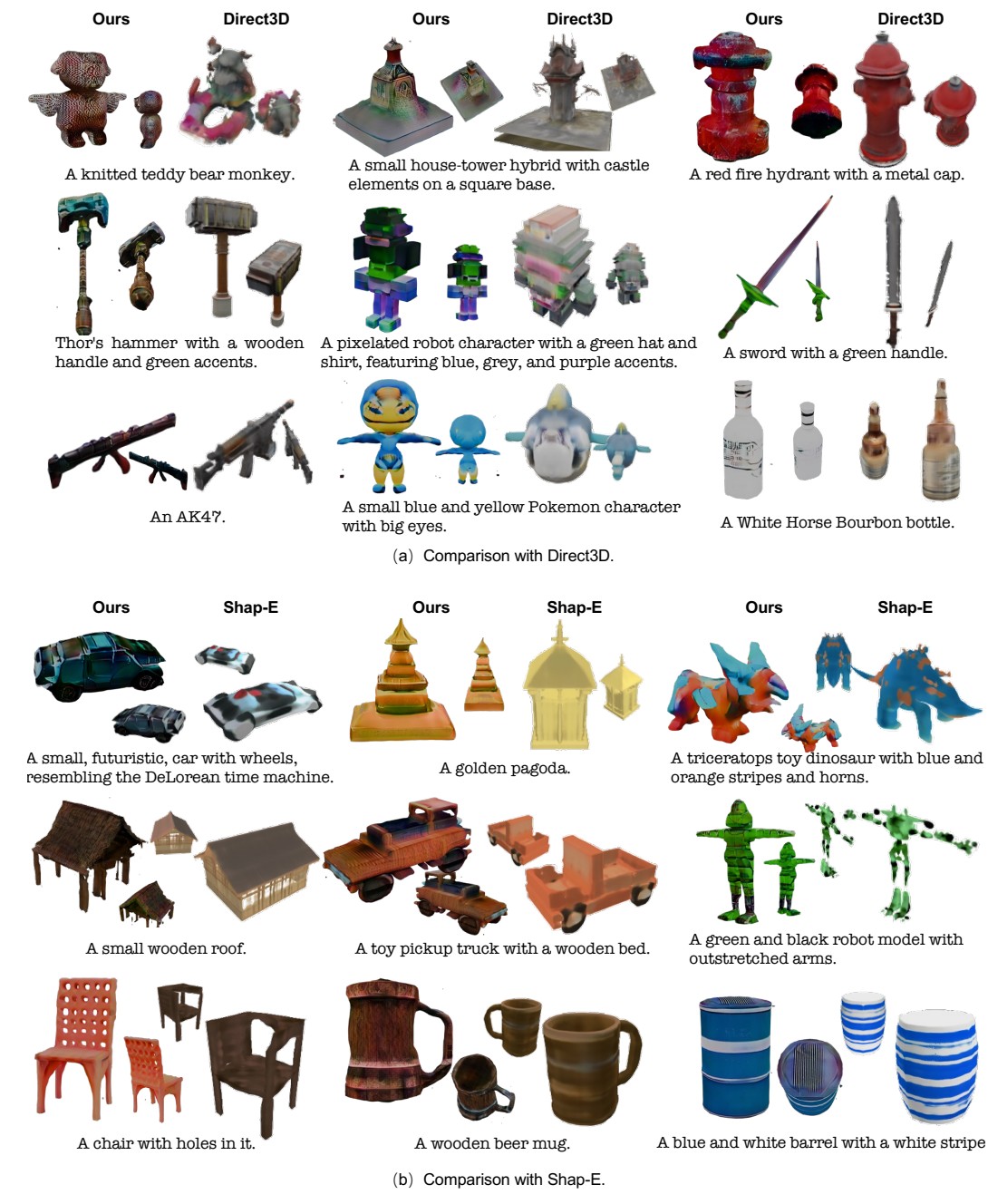

Figure 6: Comparison of generated 3D objects (without SDS refinement).

| | Geometry↑ | Texture↑ | Realistic Rendering↑ | Prompt Consistency↑ | Overall↑ |
|---|---|---|---|---|---|
| Shap-E | 3.080 | 3.019 | 2.947 | 3.121 | 3.042 |
| Direct3D | 3.222 | 3.150 | 3.118 | 3.242 | 3.183 |
| Ours | 3.456 | 3.383 | 3.332 | 3.520 | 3.423 |

Table 1: User study on generated 3D objects (without SDS refinement).

|                        | Shap-E | Direct3D | Ours   |
|------------------------|--------|----------|--------|
| openai:ViT-L-14        | 0.2398 | 0.2152   | 0.2456 |
| openai:ViT-L-14-336    | 0.2426 | 0.2220   | 0.2462 |

Table 2: CLIP score(↑) for evaluation of similarity between rendered images and given text prompt.

|                  | More Realistic (%) | More Detailed (%) | Overall (%) |
|------------------|--------------------|-------------------|-------------|
| GaussianDreamer  | 35.0               | 50.0              | 42.5        |
| GSGEN            | 17.5               | 5.0               | 11.3        |
| Ours             | 47.5               | 45.0              | 46.3        |

Table 3: User study on generated 3D objects (with SDS refinement).

GaussianDreamer (Yi et al., 2024) are selected as our baselines. The generated samples, as illustrated in Fig. 7, highlight that with the SDS refiner, our DirectTriGS can deliver competitive outcomes comparable to the state-of-the-art 2D-lifting methods.

Furthermore, we conducted a user study for quantitative evaluation. 40 users were tasked with ranking 15 samples from different methods based on two criteria: realism and level of detail. The results indicate that our approach marginally outperforms the other baseline methods.

### 5.4 ABLATION STUDY

**3D Diffusion for voxel-based Gaussian Splatting.** We attempt to do GS attribute generation conditioned on given voxel occupancy via 3D diffusion model. However, even a toy experiment of over-fitting one single object failed, which may attributes to the complex multiple channels with different distributions especially the non-Euclidean ones such as quaternion. As shown in Fig.8, the diffusion model learns to generate color but fails to generate splat scaling, opacity and orientation.

**Triplane Diffusion without VAE.** Since VAE inherently involves a reconstruction loss, we attempt to use direct diffusion on the triplane space for generation. The experiment results shows that such direct diffusion may cause serious noise on decoded geometry. Randomly generated samples are visualized in Fig.9. A possible reason is that the multi-channels of triplane contains considerable redundancy or noise, which is difficult to be captured or filtered by the diffusion model.

**Inference Efficiency.** The inference efficiency is listed in Table 4. This experiment is conducted on the platform equipped with RTX3090 GPU with 24GB memory.

## 6 CONCLUSION

We have presented a novel framework DirectTriGS for Gaussian Splatting Field generation. Direct-TriGS mainly consists of 3 parts: 1) a light-weight triplane representation for 3D object with the format of Gaussian Splatting; 2) a fully differentiable TriRenderer which can decode triplane to orginal GS and render it to multi-view images; 3) the triplane VAE and staged diffusion model for the whole generation process. By utilizing our DirectTriGS, the intricate GS data can be generated directly and efficiently. 4) Additionally, we incorporate a SDS refiner to further improve the texture and details of generated objects.

Table 4: Inference efficiency of the generation (single stage of LDM), with the batch size is 1.

|            | TFLOPs  | Parameters(M) | GPU Memory(GB) | Running Time (Second) |
|------------|---------|---------------|----------------|-----------------------|
| Generation | 0.00823 | 56.65         | 2.94           | < 8.0                 |

486
487
488
489
490
491
492
493
494
495
496
497
498
499
500
501
502
503
504
505
506
507
508
509
510
511
512
513
514
515
516
517
518
519
520
521
522
523
524
525
526
527
528
529
530
531
532
533
534
535
536
537
538
539

**Ours(Before SDS)**  **Ours(with SDS)**  **GSGEN**  **GaussianDreamer**

An astronaut in space suit.

A plane.

A green and white vase with a Greek design and logo, resembling a tea cup.

A Spiderman character in a red, yellow, and blue outfit.

A human skull.

Red and black clown mask with horns and deer head features.

Figure 7: Refinement of unsatisfactory results by SDS, compared with pure SDS-based methods GSGEN and GaussianDreamer.

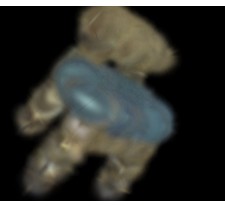

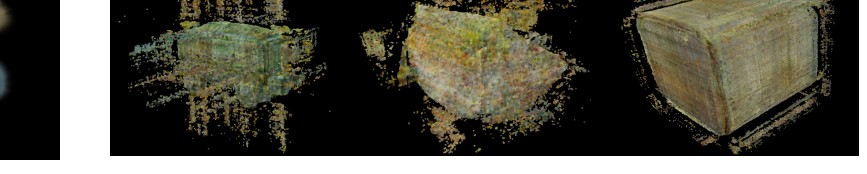

Figure 8: Failure of GS attribute generation.

Figure 9: Noisy results of direct diffusion on triplane. The condition prompt are "train", "watermelon" and "box".

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

# A APPENDIX

## A.1 DATASET INFORMATION.

Objaverse Deitke et al. (2023) is the main dataset for our experiment, which contains over 800K 3D objects. As the rendering process on such a massive dataset is very time-consuming, we adopt the pre-processed version sourced from the repository of Qiu et al. (2023), which pre-filters over 260K samples. In this processed dataset, every object is normalized to the voxel range of $[\pm 0.5, \pm 0.5, \pm 0.5]$, and rendered to RGBA images in a resolution of $512 * 512 * 4$, with 40 views in total. Our training data only comprises multi-view images and their corresponding camera poses, without any kind of original 3D data.

## A.2 IMPLEMENTATION DETAILS.

**Triplane.** The triplane resolution is configured as $3 \times 128 \times 128 \times 16$, where 16 represents the channels within each grid. The first half of the channels is designated for encoding geometry information, while the remaining half is allocated for encoding GS appearance details. Each triplane is initialized to random Gaussian noise with a standard deviation of 0.01. This random initialization allows the triplane to be decoded into random SDF values, subsequently leading to the generation of diverse fragmented mesh faces. Upon rasterization of these faces onto the screen, the geometry loss facilitates swift removal of undesired faces. During our experiments, we observed that this initialization method enables faster convergence compared to zero initialization.

As for loss configuration, we configure $w_1 = 5.0$, $w_2 = 1.0$, $w_3 = 1.2$, $\beta = 0.2$, $\gamma_1 = 0.2$, $\gamma_2 = 0.1$, $\gamma_3 = 0.01$, $\gamma_4 = 1.0$ by experiments, corresponding to the loss function described in Eq. 3, Eq. 4 and Eq. 5.

**TriRenderer.** As for the TriRenderer instroduced in Fig.1, both the geometry decoder and the GS attribute decoder inside it are composed of linear blocks. In the GS attribute decoder, there are 3 headers for GS splats scaling, opacity and SH prediction, and the rotation is fixed by the mesh face normal as introduced in Section 4.2. All the GS attribute headers are linear layers. We set SH degree to 1 in all experiments, which is enough to obtain satisfying results on Objaverse.

## A.3 COMPARISON WITH GAUSSIANCUBE.

GaussianCube (Zhang et al., 2024) is the most recent paper aiming to solve a similar task of ours, which can generate 3D GS directly without SDS or reconstruction from images. As for now, the authors of GaussianCube have not release their pre-trained models for text-to-3D task on Objaverse dataset. Therefore, we just use the images provided in their paper for a qualitative comparison. The generated samples are shown in Fig. 10. Our method produces more diverse and detailing generation results.

## A.4 MORE GENERATION RESULTS (WITHOUT SDS REFINEMENT).

More generated samples are rendered as Fig. 11.

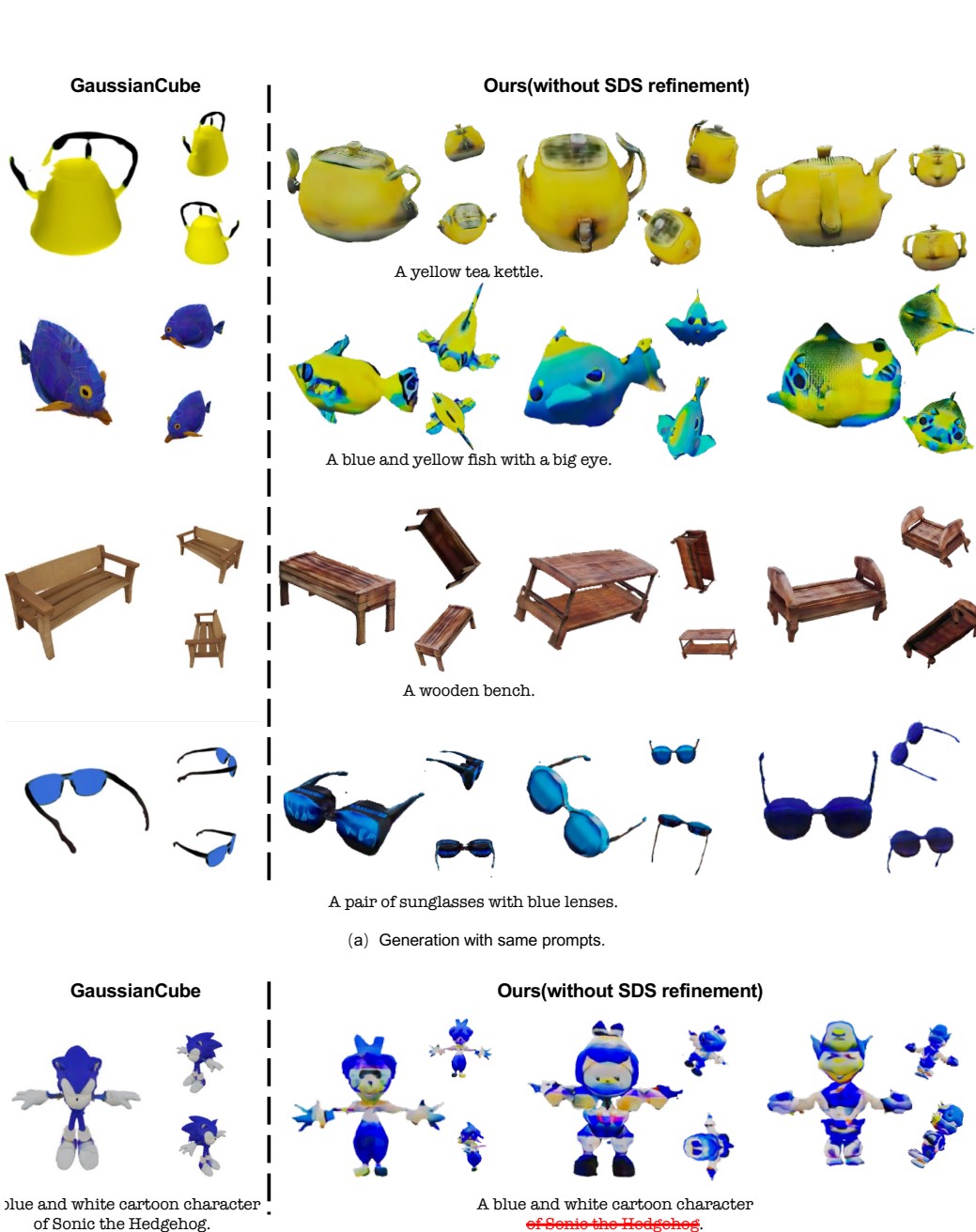

Figure 10: Comparison with GaussianCube.

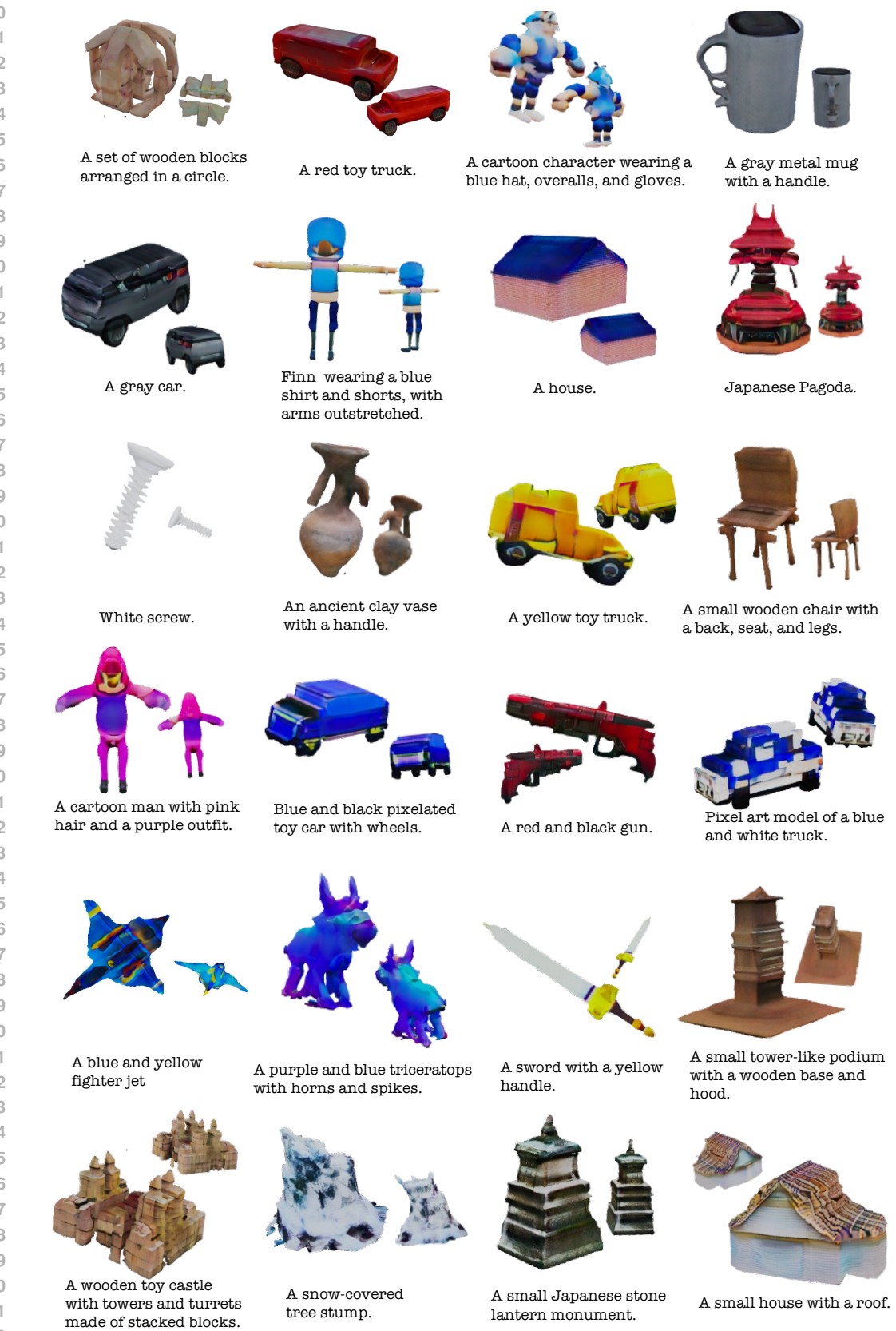

Figure 11: More generated samples (without SDS Refinement).

