# OpenReview forum: "DirectTriGS: Triplane-based Gaussian Splatting Field Representation for 3D Generation"
_ICLR.cc/2025/Conference — ICLR 2025 Conference Withdrawn Submission_

### Official Review · Reviewer_XRFH · 2024-10-29

**Soundness:** 3
**Presentation:** 3
**Contribution:** 3
**Rating:** 5
**Confidence:** 5

**Summary:**

The paper introduces **DirectTriGS**, a framework for 3D object generation using Gaussian Splatting (GS), an approach gaining traction in 3D content rendering. Traditional generative models have rarely explored directly generating 3D Gaussians due to the complex, multi-channel structure of GS data, typically represented as point clouds. DirectTriGS tackles this by using a **triplane representation** that encodes Gaussian Splatting as a continuous image-like field. This approach captures both geometry and texture information, enabling easy conversion back to Gaussian point clouds and rendering via a **differentiable renderer (TriRenderer)** with only 2D supervision. The framework leverages a **Variational Autoencoder (VAE)** to compress the triplane representation, which supports 3D object generation through latent diffusion. Experiments show that DirectTriGS achieves high-quality 3D geometry and rendering.

**Strengths:**

1. This paper presents a new framework for addressing 3D generation through the native 3D diffusion way, unlike existing (multi-view) reconstruction-based 3D generation methods.
2. The overall presentation is sound, the workload is high, and the performance is better compared to the baselines.
3. The staged VAE/latent diffusion supports more flexible 3D generation and control.

**Weaknesses:**

1. Though the overall performance is good, only text-to-3D results are shown inthe paper, where the most useful performance on i23d is not included. Since the diffusion model is agnostic to the condition, it would be great to show the performance on image-conditioned 3D generation.
2. Lack of comparison and discussions to the relevant methods. For example, the overfitting triplane - training VAE - diffusion learning pipeline is well formulated in 3D Topia. Besides, LN3Diff (ECCV 24') also adopts latent tri-plane as the intermediate representation of the 3D VAE in the diffusion training, and text-conditioned 3D generation is performed. Consider these methods are all publicly available (open sourced) before the deadline of ICLR, the comparisons are strongly requested. Besides, the similar diffusion pipeline is also well discussed in Rodin (CVPR 23), where this paper missed in the citation / discussions.
3. From the experiments in the paper, the final visual quality largely comes from the SDS fine-tuning stage, where the shape-E and Direct3D lacks. Therefore, the quantitative comparison in the table is somewhat misleading, where the raw diffusion output shall be reported rather than the fine-tuned results.
4. The visual demonstration (figures layout, experiments organization) can be greatly improved in the later version. E.g., why the fig. 6 has white background and fig. 7 / 8 switch to the black? These are not novelty issues but can be further polished.

**Questions:**

1. The design of the proposed method looks unnecessary. I understand 3DGS has many merits, but your VAE (TriRenderer pipeline) involves outputing a differentiable mesh with marching cube, where the points are sampled on the fly. Why is 3DGS necessary when you already have high-quality surface/mesh? If textures are needed, an RGB field can be optimized together like in InstantMesh / CRM. The triplane -> SDF -> 3DGS pipeline looks very wired to me, and a more straightforward design is feasible.
2. For representing 3DGS, why not directly leverage sparse point clouds and use a decoder to up-sample to the high-resolution 3DGS? This would leads to a more unified VAE pipeline with a single branch.
3. When applying the VAE of the proposed method to new 3D assets, does it require 3D reconstruction again to encode it into the latent space, or a single forward pass is good enough.
3. Regarding the ablation in Fig. 8 and Fig. 9, since there are existing diffusion methods that work on voxels and triplane, I wonder why the results shown here fail to converge?
54 Why is GaussianCube comparison not included in the main paper, but only in the appendix? It is an important baseline.

Overall, I appreciate the workload of this pipeline but it really requires more polishment. I would consider improving my rating after the author adds the required comparisons and resolves my other concerns.

---

### Official Review · Reviewer_dDoT · 2024-11-01

**Soundness:** 3
**Presentation:** 3
**Contribution:** 2
**Rating:** 5
**Confidence:** 4

**Summary:**

The paper proposes a new framework for 3D generation using triplane-based Gaussian Splatting representation. By binding the GS features to a SDF surface, the method can compress a 3D object into a triplane representation, which can be further compressed with VAE and generated using diffusion models. The TriGS representation is also fully differentiable, and can be supervised just using mutli-view images. To further enhance quality, SDS-based refinement can be applied to the generated GS. Experiments demonstrate the performance of text-to-3D generation.

**Strengths:**

* The paper is clearly written and well structured.
* Implementation details are provided for better understanding and reproduction.

**Weaknesses:**

* A major concern is that the proposed TriGS representation lacks clear motivation and advantages. It still seems to be a blunt combination of triplane-based representation and GS. For example, why do we have to use GS? It's actually unnatural to use triplane (which is continuous) for GS encoding. The model already predict a mesh, which can also be efficiently differentiable-rendered for supervision. I don't see the advantage of converting it back to point cloud and using GS to render it.
* Many related works are not properly referenced and discussed. There are many works applying triplane diffusion for 3D generation (e.g., DiffRF, Rodin, 3DTopia, ...), but not referenced or discussed in this paper. Also, the initialization of GS is similar to 2D GS or Gaussian surfels, but both are not referenced.
* The experiment results are also not very convincing in terms of quality, even with SDS refinement. With more efficient TriGS, the resolution of generated 3D objects seems to be low and not detailed, this also weakens the motivation.

[1] DiffRF: Rendering-guided 3D Radiance Field Diffusion
[2] Rodin: A generative model for sculpting 3d digital avatars using diffusion
[3] 3DTopia: Large Text-to-3D Generation Model with Hybrid Diffusion Priors
[4] 2D Gaussian Splatting for Geometrically Accurate Radiance Fields
[5] High-quality Surface Reconstruction using Gaussian Surfels

**Questions:**

* How many points are sampled from the mesh surface? What is the average number of Gaussians for the generated objects?
* For the two-staged training, how to make sure the TriRenderer is good enough for generalization? How to choose the 1000 objects subset?
* There are many losses during training triGS and VAE. It would still be better to perform some ablation. For example, the weight of KL loss may be crucial to balance the reconstruction quality and latent space smoothness. But the paper only says a "small" weight without further details.

---

### Official Review · Reviewer_7yn1 · 2024-11-02

**Soundness:** 3
**Presentation:** 3
**Contribution:** 2
**Rating:** 5
**Confidence:** 4

**Summary:**

This paper proposes a diffusion-based text-to-3D generation method to generate 3D Gaussian Splatting models. The framework has four key components: Triplane-based 3D representation, Triplane VAE, LDM latent generation and SDS refinement. The main contribution lies in the interesting Triplane representation paired with TriRenderer, which has SDF branch to extract geometry and GS branch to generate textures. The standard LDM approach is used for text-to-3D generation. Experiments prove that the proposed method generates better results compared with existing text-to-gaussian approaches.

**Strengths:**

1. This paper is well written and easy to follow.
2. I appreciate the interesting TriRender and Triplane-based 3D GS representations, especially the requirement of only multi-view images training.
3. Experiments validate the proposed method outperforms existing text-to-3DGS approaches.

**Weaknesses:**

1. The quality of the results is not good enough. As shown in Figure 6 and 11, the generated 3D models have strange geometry distortion and fuzzy texture details. The inadequate quality of the 3D models makes me negative about this submission.
2. The comparison should be further improved. There are many NeRF-based SDS generation approaches, such as MVDream and DreamCraft3D. These approaches have generated high quality results, which seem to largely outperform the results in Figure 6 and 11. Please add more comparisons and discussions.
3. In Figure 10, it is hard to conclude that the proposed method generates better results than GaussianCube, especially the noisy geometry and texture exhibited in the proposed method.

**Questions:**

1. Text-to-3D generation always suffers from the overfitting to the text prompts. Are the results shown in paper all generated with text prompts in validation dataset?
2. This paper solves the text-to-3D generation problem. However, image-to-3D is a more popular problem with many successful approaches such as Wonder3D and Triplane-Meet-Gaussian. What are the advantages of the text-to-3D compared with image-to-3D? Text-to-3D can also be solved with 2D image generation combined with image-to-3D approach.
3. In TriRender, the point cloud is generated by the surface point sampler from mesh. Is this sampler differentiable? Please add more discussion about it.
4. For the quantitative experiments, were the same 50 samples simultaneously used for user study and CLIP evaluation?

---

### Official Review · Reviewer_q1Du · 2024-11-03

**Soundness:** 3
**Presentation:** 3
**Contribution:** 2
**Rating:** 5
**Confidence:** 5

**Summary:**

The paper introduces DirectTriGS, a framework for 3D object generation leveraging Gaussian Splatting (GS). The authors address the challenge of complex data structures in traditional GS by proposing a triplane representation, which enables the encoding of both geometric and textural information into a continuous field. This representation facilitates the transformation back to Gaussian point clouds and rendering into images with only 2D supervisions. The framework includes a fully differentiable TriRenderer for end-to-end training and a Variational Autoencoder (VAE) for compression, which is then utilized in latent diffusion for 3D object generation. Experiments demonstrate high-quality 3D object geometry and rendering results.

**Strengths:**

1. The novel point: The framework includes a fully differentiable TriRenderer and utilizes VAEs and latent diffusion, which are state-of-the-art techniques in the field.
2. The paper provides thorough experiments and comparisons with existing methods, demonstrating the effectiveness of the proposed approach. The generated 3D object geometries and renderings are of high quality, indicating the potential of DirectTriGS for practical applications, but still lack some high-frequency appearance details. Also, some relative research work is not considered for comparison, such as BrightDreamer, Align Your Gaussians, and Triplane Meets GaussianSplatting; they all use the triplane into gaussian splatting for shape generation.
3. The paper is well organized, with a clear presentation of the methodology, experiments, and results.

**Weaknesses:**

1. While the paper introduces an approach, the complexity of the triplane representation and the need for a fully differentiable renderer may make the framework challenging for some researchers to implement. A simplified version or more detailed implementation guidance could be beneficial.
2. For table 4, what is the meaning to list the inference efficiency of a single stage of LDM only? Since the pipeline uses the SDS optimization, the efficiency of whole pipeline is not higher, right? The paper could provide more details on the computational efficiency of DirectTriGS of each step, such as training and inference times, especially compared to other methods.
3. The paper could benefit from a more extensive user study to evaluate the generated 3D objects from different perspectives, such as usability, realism, and preference. From the results, the performance of the proposed method cannot beat GaussianDreamer fully.
4. The recent works [1, 2, 3] have a similar core idea; the comparison should be conducted for a total evaluation of the proposed methods.
[1] BrightDreamer: Generic 3D Gaussian Generative Framework for Fast Text-to-3D Synthesis
[2] Align Your Gaussians: Text-to-4D with Dynamic 3D Gaussians and Composed Diffusion Models
[3] Triplane Meets Gaussian Splatting: Fast and Generalizable Single-View 3D Reconstruction with Transformers
5. While the paper mentions an ablation study, more detailed analysis on the contribution of each component of the framework could strengthen the claims. BTW, Since there are more loss terms, how to balance the weight during the optimization and the effects should be fully evaluated in the ablation studies.
6. In figure 1, the necessary detials are not provided so that fully understand the core idea of the proposed methods. i.e. what is the necessity of the deformable SDF volume, deform what? Why not directly optimize the Guasisan points position, instead of the first generating geometry and synthesis the GS attributes secondly. And for the GS Attribute decoder, does the position of the sampled points to be optimized. Does the triplane feature extractor be the same for the different branch?  Since the input of two triplane feature extractor is different.

**Questions:**

The paper is well-written and presents a novel point to the field of 3D generation. Based on the above comments: some major concerns, the insufficient evaluations (i.e., the comparison with recent work and ablation studies on heavy loss terms), weak performance from the user study, uncleared figure presentation. With the above suggestions addressed, the paper would be a strong candidate for publication. I recommend a borderline score and am negative for the insufficient evaluations.
Detailed questions refer to weaknesses.

---

### Note · Authors · 2024-11-14

I have read and agree with the venue's withdrawal policy on behalf of myself and my co-authors.